# POINT NEIGHBORHOOD EMBEDDINGS

## ABSTRACT

Point convolution operations rely on different embedding mechanisms to encode the neighborhood information of each point in order to detect patterns in 3D space. However, as convolutions are usually evaluated as a whole, not much work has been done to investigate which is the ideal mechanism to encode such neighborhood information. In this paper, we provide the first extensive study that analyzes such Point Neighborhood Embeddings (PNE) alone in a controlled experimental setup. From our experiments, we derive a set of recommendations for PNE that can help to improve future designs of neural network architectures for point clouds. Our most surprising finding shows that the most commonly used embedding based on a *Multi-layer Perceptron* (MLP) with ReLU activation functions provides the lowest performance among all embeddings, even being surpassed on some tasks by a simple linear combination of the point coordinates. Additionally, we show that a neural network architecture using simple convolutions based on such embeddings is able to achieve state-of-the-art results on several tasks, outperforming recent and more complex operations. Lastly, we show that these findings extrapolate to other more complex convolution operations, where we show how following our recommendations we are able to improve recent state-of-the-art architectures.

## 1 INTRODUCTION

In computer vision, point clouds are one of the most commonly used representations to process and store 3D data. This is because point clouds are a compact representation and most 3D acquisition hardware produces point clouds as their output. In the past years, the advances in 3D acquisition hardware, and therefore the number of available point clouds, allowed the development of data-driven methods to solve different 3D scene understanding problems. In particular, the pioneering work of Qi et al. (2017a) opened the door to new neural network architectures which were able to process point clouds directly. Since then, researchers have developed several convolution operations for point clouds trying to mimic the success of *Convolutional Neural Networks* (CNN) for images (Thomas et al., 2019; Mao et al., 2019; Atzmon et al., 2018; Hermosilla et al., 2018; Lin et al., 2022).

The most commonly used convolution operations for point clouds can be divided into two main groups based on the embedding function used to encode the neighborhood information of each point: *Kernel Points* (KP) embeddings and MLP-based embeddings. The KP-based embeddings define a set of points in the receptive field and a correlation function to compute the embedding of each neighboring point (Thomas et al., 2019; Atzmon et al., 2018; Mao et al., 2019). MLP-based embeddings, on the other hand, use an MLP to learn the embedding function (Hermosilla et al., 2018; Wu et al., 2019; Qian et al., 2022; Lin et al., 2022). However, previous work has presented such PNE as part of a convolution operation and little attention has been given to which of these embedding mechanisms improve learning and what are desirable properties for such embeddings.

To fill this gap, in this work, we present the first extensive study in a controlled experimental setup that analyzes the performance of different PNE on several downstream tasks. From our experiments, we derive a set of best practices to improve PNE design that will help the development of future neural network architectures. Among our findings, we show that, in general, KP embeddings provide better performance than commonly used MLP embeddings. Surprisingly, our experiments show that commonly used MLP-based embeddings with ReLU (Fukushima, 1975) activation functions provide the lowest performance of all, and in some cases do not provide an improvement over the use of the point coordinates directly. Moreover, we show that the neighborhood selection mechanism plays a crucial role in the final performance of the model, where a *Ball-query* (BQ) method is more stable

than the commonly used *k-Nearest Neighbors* (kNN). We also show how a neural network using such embeddings in a simple convolution operation can achieve state-of-the-art results on several tasks, improving over recent and more complex convolution operations and architectures. Lastly, we show how our recommendations can be used to modify existing convolution operations, leading to better performance and more stable training. The code will be made publicly available upon acceptance.

## 2 STATE OF THE ART

Neural network architectures to directly process unstructured point clouds have been an active area of research in the last few years. The groundbreaking work of Qi et al. (2017a) proposed the PointNet architecture. This neural network processed each point coordinate independently by an MLP which resulting features were then aggregated over the whole point cloud to perform a final prediction. Later, the same authors proposed an extension of such architecture, PointNet++ (Qi et al., 2017b), in which they used similar ideas but in a hierarchical design, imitating conventional CNNs for images. After, several authors followed up on these works and proposed more advanced neural network architectures: Atzmon et al. (2018), Thomas et al. (2019), Boulch (2020), and Mao et al. (2019) proposed convolution operations based on KP embeddings. On the other hand, Hermosilla et al. (2018) and Wu et al. (2019) investigated operations where the embedding is learned with an MLP.

Despite the efforts to develop a successful convolution operation for point clouds, commonly used benchmarks were dominated by sparse discrete convolutions (Graham et al., 2018; Choy et al., 2019). However, recent research (Qian et al., 2022; Lin et al., 2022) has shown that the improvements brought by such methods came from the data augmentation and training strategies rather than the convolution operation. These works proposed simple MLP-based convolutions that are able to achieve new state-of-the-art results on several data sets. Moreover, in a similar line of research, several authors have proposed MLP-based convolution operations using different self-attention mechanisms (Zhao et al., 2021; Wu et al., 2022; 2023) which have been shown also to outperform sparse convolutions.

## 3 BACKGROUND: CONVOLUTION OPERATION

In the core of most neural network architectures for point clouds, there is a convolution operation that detects patterns at each point location. In this paper, we use a general definition of convolution operation, that encompasses several existing works.

$$F^l(\mathbf{x}) = \sum_{c=0}^{I} \sum_{\mathbf{y} \in \mathcal{N}(\mathbf{x})} F_c^{l-1}(\mathbf{y}) \kappa_c(e(\mathbf{y} - \mathbf{x})) \tag{1}$$

where $F_c^l(\mathbf{x})$ is the feature $c$ of layer $l$ in position $\mathbf{x}$, $\mathcal{N}(\mathbf{x})$ is the set of neighboring points of $\mathbf{x}$, $\mathbf{y}$ is a neighboring point of $\mathbf{x}$, $\kappa$ is the learnable continuous kernel, and $e$ is the embedding function.

Note that in this definition, $\mathbf{x}$ and $\mathbf{y}$ do not have to belong to the same point cloud, which allows using the convolution to transfer features between different point clouds. Also, note that this definition of convolution does not capture all existing convolution operations such as recent attention-based operations (Zhao et al., 2021; Lai et al., 2022; Wu et al., 2022). To incorporate such architectures, the input to the embedding $e$ should be updated to take the features of points $x$ and $y$ instead. However, we use this simple framework to analyze the embedding functions, and, as we will show later, the insights gained from this work can be used to improve embeddings on more complex operations.

**Neighborhood** Neighborhood selection is a design decision that can significantly impact the performance of the final convolution operation independently of the neighborhood embedding used. Moreover, different embeddings might perform differently based on the neighborhood selection. Several methods (Wu et al., 2019; Zhao et al., 2021) have used kNN to select the neighboring points of a given position $\mathbf{x}$. This method simplifies the code design in common learning frameworks that work with tensors since it ensures that each point has the same number of neighbors. However, this method can generate receptive fields with different sizes for two locations in space due to variable point densities (Hermosilla et al., 2018; Qian et al., 2022). Another commonly used approach is to use ball-query (Hermosilla et al., 2018; Thomas et al., 2019; Qian et al., 2022; Wu et al., 2022) to select all points at a distance $r$ to the query point $\mathbf{x}$. Contrary to kNN, this approach has a fixed receptive field

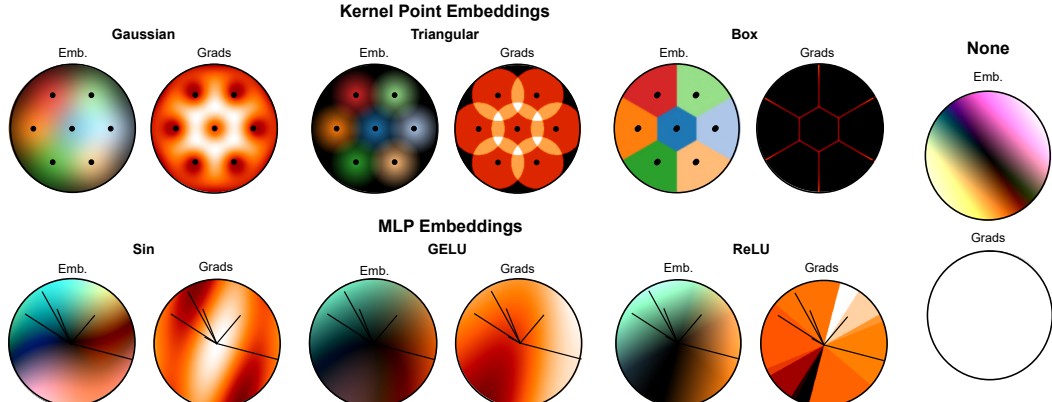

Figure 1: Different point neighborhood embeddings visualized in 2D. Colors represent the absolute value in each embedding dimension. Next to each embedding, we illustrate the gradient norm. **Top:** Point neighborhood embeddings based on kernel points with different correlation functions, *Box*, *Triangular*, and *Gaussian* functions. **Bottom:** Point neighborhood embeddings based on MLP with different activation functions, *ReLU*, *GELU*, and *Sin* activation functions. Moreover, the axis direction of each dimension is represented with a line. **Right:** Direct 3D coordinates are used as embedding.

independent of the point density. However, it makes the implementation of convolutions cumbersome due to the variable number of neighbors. In this work, we test all PNE with both approaches.

**Embedding function**    In this work, we define the neighborhood embedding function $e$ as a function of the relative neighbor position $y - x$, $e : \mathbb{R}^3 \to \mathbb{R}^E$, where $E$ is the dimension of the embedding. This function can be any type of function that captures the shape of the neighborhood and helps train the kernel, *e.g.* identity function, point correlations, or can even be learned by a small MLP.

**Kernel**    We define our kernel as a function that takes as input the embedding dimensions of each neighbor and predicts the kernel value used in the convolution operation, $\kappa : \mathbb{R}^E \to \mathbb{R}^1$. This function is usually learned and the one performing the detection of patterns. Multiple definitions of $\kappa$ exist, but, here we use a simple linear combination of the input embedding dimensions. When put together for all the input and output features of a layer, the kernel is learnable tensor $\kappa \in \mathbb{R}^{I \times O \times E}$, where $I$ is the number of input features, $O$ is the number of output features, and $E$ is the embedding size.

## 4    POINT NEIGHBORHOOD EMBEDDINGS

In this section, we discuss the most commonly used neighborhood embedding mechanisms in point convolutional neural networks and new embedding mechanisms derived from other architectures.

### 4.1    KERNEL POINT EMBEDDINGS

Several authors have suggested using as embedding the correlation to a set of kernel points placed on the receptive field (Thomas et al., 2019; Atzmon et al., 2018; Mao et al., 2019). These methods can be structured along two axes: correlation function used and position of kernel points.

#### 4.1.1    CORRELATION FUNCTION

The correlation function of a KP embedding is a function that defines the correlation between the neighboring point and the kernel points. Next, we analyze existing correlation functions.

**Box.**    The correlation function most similar to discrete convolutions is the box function. This function has been used in the past in cartesian (Hua et al., 2018) or in spherical coordinates (Lei et al., 2019) to encode the neighborhood information. More recently, this approach has been used in discrete 3D transformer architectures to encode the relative position of points within a voxel (Lai et al., 2022).

An illustration of this correlation function and its gradients w.r.t. the point coordinates is depicted in Fig. 1. When analyzed, we can see that each function used to compute the dimensions of the embedding has independent support, which results in a point embedding similar to one-hot encoding where only one dimension has a value equal to one and the rest are zeros. Moreover, we can see that the support of the embedding is equal to the receptive field, i.e. all points inside of the receptive field have an embedding with a norm higher than zero. We can also see that this embedding function is not continuous. Lastly, with this embedding, two different points can have the same embedding. When looking at the gradients, we can see that point coordinates have zero gradients everywhere in the receptive field, which might make this embedding not suited for tasks where gradients for point coordinates are required, e.g. generative models.

**Triangular.** Thomas et al. (2019) proposed a triangular function as correlation function, defined as:

$$e_j(\mathbf{x}) = max\left(1 - \frac{\|\mathbf{p}_j - \mathbf{x}\|}{\sigma}, 0\right) \tag{2}$$

where $\mathbf{p}_j$ is the $j$ kernel point, and $\sigma$ is a parameter that controls the extent of the embedding function. See Fig. 1 for an illustration. This embedding, contrary to the *Box* embedding, is continuous. A variation of a point coordinate results in most cases on a different embedding. Moreover, with the correct $\sigma$, the support of the embedding is equal to the receptive field. When looking at the gradients, we can see that gradients w.r.t to point coordinates are higher than zero. However, we can see that this embedding function is not differentiable due to the discontinuities introduced by the $max$ function.

**Gaussian.** Several authors have suggested to use Gaussian functions as a correlation function (Atzmon et al., 2018; Mao et al., 2019). This approach uses a Gaussian centered at each kernel point:

$$e_j(\mathbf{x}) = e^{\frac{-\|\mathbf{p}_j - \mathbf{x}\|^2}{2\sigma^2}} \tag{3}$$

where $\sigma$ controls the extent of the function. This embedding has similar properties as the *Triangular* embedding. However, their main difference is that this embedding is differentiable, see Fig. 1.

### 4.1.2 KERNEL POINT POSITIONS

Another important design choice of KP embeddings is the location of the kernel points. In the following paragraphs, we discuss existing choices of kernel points arrangement.

**Regular grid.** Most of the works that rely on *Box* (Hua et al., 2018; Lei et al., 2019) and *Gaussian* (Atzmon et al., 2018; Mao et al., 2019) correlation functions proposed to use kernel point positions following a regular grid structure. This setup is a natural evolution of discrete convolutions, where the space is discretized into voxels. However, for continuous point positions and *Radial Basis Function* (RBF) such as *Gaussian* or *Triangular* functions, all areas might not be part of the support. The accompanying figure illustrates this in 2D. Note that this effect increases in higher dimensions.

**Platonic solids.** Thomas et al. (2019) suggested optimizing the point positions in a pre-processing step to guarantee equal coverage of the receptive field. This process resulted in the vertex positions of platonic solids for a specific number of points. The accompanying figure presents the kernel point arrangement following the vertices of an icosahedron. This kernel point arrangement is more suited for continuous point coordinates since the space is equally covered by the correlation functions and does not suffer from the same problems of a regular grid placement.

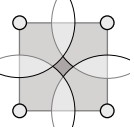

**Adaptive locations.** Thomas et al. (2019) also proposed to learn these kernel point positions from a previous convolution operation with fixed kernel point positions. However, this method requires additional regularization losses to push the kernel point locations to the surface of the patch.

### 4.2 MULTI-LAYER PERCEPTRON EMBEDDINGS

One of the most commonly used PNE is to use a single-layer MLP followed by an activation function (Hermosilla et al., 2018; Wu et al., 2019; Lin et al., 2022; Wu et al., 2022; 2023):

$$e(\mathbf{x}) = \alpha(\mathbf{W}\mathbf{x} + \mathbf{b}) \tag{4}$$

where $\alpha$ is the activation function. This choice of embedding function is due to the universal approximation properties of such networks. Most of existing works have used a ReLU (Fukushima, 1975) activation function as their choice of $\alpha$. In the following paragraphs, we describe this choice of activation function and propose to use two additional activation functions.

**ReLU.** The ReLU activation function (Fukushima, 1975) clamps negative values to zero. Fig. 1 illustrates the resulting embeddings and their gradients. We can see that the support of embedding dimensions overlaps, which can make two dimensions of the embedding redundant. Moreover, the support of the embedding might not be equal to the receptive field, as there can be areas for which all embedding dimensions are equal to zero. When looking at the gradients, we can see that this embedding is not differentiable with zero-gradient areas, which might make learning $\mathbf{W}$ difficult.

**GELU.** Extensive research efforts have been devoted to finding a differentiable version of the ReLU activation function (Clevert et al., 2016; Hendrycks & Gimpel, 2016). Recent state-of-the-art models in different areas of computer vision (et al, 2021b;a; 2022b;a) have switched ReLU activation functions (Fukushima, 1975) by GELU activation functions (Hendrycks & Gimpel, 2016), leading to improved performance on several tasks. In this paper, we propose to use GELU activation functions on MLP-based PNE. This activation function is defined as $x\Phi(x)$ where $\Phi(x)$ is the Gaussian cumulative distribution function. Fig. 1 illustrates the resulting embeddings and their gradients. We can see that we obtain almost the same embedding as the one obtained with the ReLU activation function. However, when we look at the gradients we can see that not only the embedding is differentiable on the whole receptive field but we also have gradients higher than zero almost everywhere.

**Sin.** Recent works on implicit representations using MLP have proposed to use trigonometric functions as activation functions (Sitzmann et al., 2020) or as the initial embedding of world positions (Tancik et al., 2020). This approach is related to the Fourier Transform where spatial coordinates are decomposed into different frequencies. Recently, Zhang et al. (2023) also suggested using such an approach for point cloud learning. Similar to these works, we use the $sin$ function as a possible activation function, see Fig. 1. We can see that the embedding is differentiable with gradients higher than zero almost everywhere in the receptive field. Compared to GELU, $sin$ activation function provides higher gradients to the central point, whereas GELU has almost zero gradients when the biases are initialized to zero. More importantly, for the $sin$ activation function, the range of the embedding is restricted to the range $[-1, 1]$, while in GELU it is unbounded for the positive axis. This might pose a problem for kNN neighborhoods with outlier points.

## 5 Embedding Evaluation

In this section, we describe the experiments carried out to evaluate the different PNE.

### 5.1 Network Architecture

In our experiments, we use an encoder-decoder architecture. The input point cloud is processed using convolutions and a set of Metaformer blocks (Yu et al., 2022) with convolutions at their core. The process is repeated for different point cloud resolutions obtained using the Cell-Average method (Thomas et al., 2019), increasing the cell size in each level by a factor of two. To transfer features between different point clouds, we use a convolution operation. For classification tasks, we perform global pooling on the last level and a linear layer performs the final predictions. For segmentation, we use a decoder with skip connections similar to Kirillov et al. (2019).

### 5.2 PNE

We evaluate three different MLP embeddings by selecting different activation functions, *ReLU*, *GELU*, and *Sin* functions. We use an embedding dimension of $E = 16$. Moreover, we evaluate three different KP embeddings by selecting different correlation functions, *Box*, *Triangular*, and *Gaussian*. As kernel points, following the motivation of kernel point placement described in SubSec. 4.1.2, we place 12 kernel points on the vertices of an icosahedron. Moreover, we create an additional kernel point on the center of the receptive field, resulting in $E = 13$ embedding dimensions. Lastly, we also

Table 1: Comparison of different PNE on the tasks of classification, ScanObjNN, and semantic segmentation, ScanNet.

| Neigh. | Emb. | Type | ScanObjNN | | ScanNet | |
|---|---|---|---|---|---|---|
| | | | Acc | mAcc | mIoU | mAcc |
| BQ | KP | Box | $92.5 \pm 0.2$ | $\underline{91.1} \pm 0.4$ | $\underline{72.7} \pm 0.3$ | $\underline{80.7} \pm 0.4$ |
| | | Trian | $\underline{92.6} \pm 0.2$ | $91.0 \pm 0.2$ | $72.4 \pm 0.5$ | $80.4 \pm 0.3$ |
| | | Gauss | $\mathbf{92.9} \pm 0.6$ | $\mathbf{91.6} \pm 0.7$ | $\mathbf{73.1} \pm 0.3$ | $\mathbf{80.8} \pm 0.2$ |
| | MLP | ReLU | $91.1 \pm 0.6$ | $89.6 \pm 0.7$ | $71.4 \pm 0.6$ | $79.4 \pm 0.7$ |
| | | GELU | $92.8 \pm 0.2$ | $91.4 \pm 0.3$ | $71.9 \pm 0.4$ | $80.0 \pm 0.4$ |
| | | Sin | $91.9 \pm 0.1$ | $90.6 \pm 0.1$ | $72.4 \pm 0.3$ | $80.1 \pm 0.4$ |
| | None | | $92.4 \pm 0.1$ | $91.1 \pm 0.1$ | $71.1 \pm 0.3$ | $78.9 \pm 0.1$ |
| KNN | KP | Box | $91.3 \pm 0.5$ | $89.9 \pm 0.8$ | $72.2 \pm 0.2$ | $\underline{80.3} \pm 0.2$ |
| | | Trian | $91.4 \pm 0.8$ | $\underline{90.0} \pm 0.7$ | $\underline{72.3} \pm 0.3$ | $79.9 \pm 0.2$ |
| | | Gauss | $91.0 \pm 0.5$ | $89.2 \pm 0.3$ | $\mathbf{72.5} \pm 0.1$ | $80.6 \pm 0.2$ |
| | MLP | ReLU | $89.9 \pm 0.4$ | $88.0 \pm 0.8$ | $71.0 \pm 0.4$ | $78.9 \pm 0.6$ |
| | | GELU | $91.0 \pm 0.7$ | $89.8 \pm 1.1$ | $71.0 \pm 0.3$ | $79.1 \pm 0.2$ |
| | | Sin | $\mathbf{92.2} \pm 0.1$ | $\mathbf{90.6} \pm 0.1$ | $72.1 \pm 0.2$ | $79.8 \pm 0.1$ |
| | None | | $90.1 \pm 0.9$ | $88.3 \pm 0.1$ | $70.4 \pm 0.2$ | $78.8 \pm 0.2$ |

evaluate not using an embedding at all, where the point coordinates are directly used as embedding dimensions, resulting in $E = 3$ embedding dimensions.

Since different embedding mechanisms have different embedding dimensions $E$, the size of the tensor representing the kernel significantly varies, $\kappa \in \mathbb{R}^{I \times O \times E}$. This makes the resulting models have different number of parameters, making them difficult to compare. Therefore, we apply a learnable matrix that transforms the resulting embedding dimension to a common embedding dimension size, $E = 16$ in our experiments, which results in an equal number of parameters for the same architecture.

Moreover, since neighborhood selection can play a crucial role in the performance of an embedding, we test all experiments with kNN and BQ neighborhood selection methods. We use $k = 16$ in kNN and, in BQ, we use as radius $r = sd$, where $d$ is the cell size of the subsampled point cloud and $s$ a scale factor that selects on average 16 neighbors over all layers ($s = 2$). Moreover, for BQ we position the kernel points at distance $.6r$ and at distance $1.2r'$ in kNN, being $r'$ the average neighbor distance over the whole training set.

## 5.3 DATASETS

We evaluate all PNE on two tasks, classification and semantic segmentation. While in classification a model can perform the task from sparsely sampled point clouds, in segmentation the model is more sensitive to small variations of point positions. We believe these two tasks are representative of common tasks in 3D computer vision. Therefore, we use the ScanObjNN (Uy et al., 2019) dataset for classification and the ScanNetV2 (Dai et al., 2017) dataset for semantic segmentation.

**ScanObjNN (Uy et al., 2019).** This dataset is composed of 3D scans of real objects for which each needs to be classified into one of 13 different classes. The dataset provides $2,315$ objects for training and $587$ for testing. The raw scans contain 3D coordinates, $[x, y, z]$, for each point, its normal, $[n_x, n_y, n_z]$, and color, $[r, g, b]$. The dataset defines three different variants of the task. OBJ_ONLY, where only points of the object are provided to the network, OBJ_BG, where points from the object and the surrounding objects are input to the model, and PB_T50_RS, where the objects are perturbed by random rotations, scaling, and translations. We evaluate different PNE on OBJ_ONLY, and compare to state-of-the-art methods on OBJ_BG and PB_T50_RS. Performance is measured with overall accuracy and per class mean accuracy.

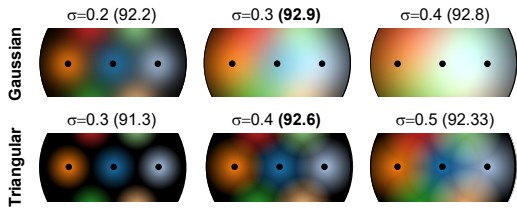

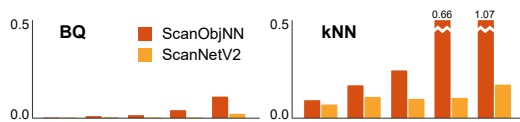

Figure 2: RBF correlation functions with different $\sigma$ and their classification performance. Small $\sigma$ makes the support smaller than the receptive field. Big $\sigma$ makes the functions overlap.

Figure 3: Variance of the normalized distance to the farthest point in each neighborhood for the different layers of the network. We can see that BQ neighborhood selection maintains a low variance even for the ScanObjNN dataset composed of single shapes. kNN, on the other hand, has a higher variance on the neighborhood selection.

**ScanNetV2 (Dai et al., 2017).** This dataset is composed of real 3D scans of $1,513$ different rooms. Among the different tasks of this benchmark, we focus on the task of semantic segmentation where the network has to predict the class of the object to which each point belongs among 20 different classes. For each point in the 3D scan, 3D coordinates, $[x, y, z]$, its normal, $[n_x, n_y, n_z]$, and color, $[r, g, b]$, are provided. The $1,513$ rooms are divided into two splits, $1,201$ rooms for training and $312$ rooms for validation. Additional 100 rooms are provided for testing where the ground truth annotation is not available. We use the validation set to compare PNE and the validation and test set to compare to other methods. Performance is measured with per class mean IoU and accuracy.

## 5.4 RESULTS

Tbl. 1 presents our main results, where the mean and standard deviation are computed over three runs.

**KP.** We can see that all KP embedding methods achieve good performance on all tasks, always improving over not having an embedding at all. Moreover, we can see that all correlation functions achieve similar performance on all tasks, with *Gaussian* correlation function being the one that obtains slightly better performance. Fig. 2 presents the results on the task of object classification when we increase/decrease the parameter $\sigma$ of the two RBF used as correlation functions, *Triangular* and *Gaussian*. Increasing $\sigma$ makes the support of the correlation functions overlap, leading to a decrease in performance. On the other hand, decreasing $\sigma$, reduces the overlap, but makes some areas of the receptive field not part of the support.

**MLP.** When we look at the results obtained by MLP-based PNE, we can see that the most commonly used embedding, using a *ReLU* activation function, achieves the worst performance of all. For classification, this method achieves even worse performance than not using an embedding at all, indicating that this embedding is not a good option for capturing the neighborhood information. However, when we apply the two activation functions suggested in this paper, *GELU* and *Sin*, we can see that the performance of the model increases, always surpassing not using an embedding. When these two methods are compared, we can see that *Sin* activation function surpasses *GELU* in almost all tasks. This difference increases significantly for kNN neighborhoods, where the restricted range of the *Sin* function is able to better cope with outlier points.

**KP vs MLP.** Results show that using a KP-based embedding on average performs better than using an MLP-based embedding. However, when using the *Sin* activation function in MLP embeddings the performance gap is reduced and in some cases even surpassing all KP embeddings.

**kNN vs BQ.** When comparing the results of PNE for different neighborhood selections, we can see that on average, BQ neighborhood selection provides a performance improvement over kNN. We believe this might be the result of the variable receptive field of kNN. This will result in a large embedding norm for some MLP-based embeddings, *ReLU* and *GELU*, and some outlier points not being part of the support for some KP-based embeddings, *Gaussian* and *Triangular*. To confirm that this variance exists, we analyze the receptive field extent for kNN and BQ neighborhood embeddings. Fig. 3 presents the variance of the distance to the farthest point in each neighborhood normalized by the cell size used for subsample each level. We can see that for BQ, the variance is low for all

layers and datasets. For kNN, we can see that this variance is higher. Moreover, we can see that for object classification, this variance significantly increases for deeper layers of the network. Point clouds in this task represent single objects and, on lower point cloud resolutions, with kNN translates to large receptive fields for some shapes. For the task of semantic segmentation, objects are usually surrounded by other objects, making the receptive field less variable. This might explain the poor performance of kNN on the ScanObjNN dataset.

## 5.5 BEST PRACTICES

In this section, we summarize a set of good practices for selecting PNE on convolution operations for points clouds: **(1)** Overall, KP-based neighborhood embeddings provide the most stable performance when compared to MLP. **(2)** Despite common practices, for KP embeddings, *Triangular* correlation functions provide slightly worse performance than *Gaussian* correlation functions. **(3)** Contrary to current trends in the field, *ReLU* activation functions should be avoided on MLP embeddings, and continuous differentiable activation functions such as *GELU* and specially *Sin* should be used instead. **(4)** As neighborhood selection method, BQ presents an advantage over kNN where the variance of the receptive field extent highly depends on the tasks at hand.

From the experiments presented in this paper, we summarize a set of best practices for designing new PNE: **(1)** The results suggest that continuous and differentiable functions provide better learning signal for the convolution operation, *ReLU* vs. *GELU* and *Triangular* vs. *Gaussian*. **(2)** Moreover, the results in Fig. 2 suggest that the support of the embedding should be equal to the receptive field. **(3)** Lastly, having a bounded range helps the embedding function to cope with outlier points in kNN.

## 5.6 COMPARISON TO STATE-OF-THE-ART

In this section, we show how a neural network architecture using as convolution a simple linear combination of KP embeddings with *Gaussian* correlation functions is able to outperform other more complex architectures. We compare our model to recenmethods on the semantic segmentation task of the ScanNet v2 dataset and on the tasks OBJ_BG and PB_T50_RS from the ScanObjNN dataset.

Following common practices in the ScanNetV2 dataset, we report mean intersection over union on the validation and test sets. Tbl. 2 shows that our model is able to outperform all standard convolutional approaches based on points such as PointNet++ (Qi et al., 2017b), PointConv (Wu et al., 2019), KPConv (Thomas et al., 2019), or PointMetaBase (Lin et al., 2022), and all standard convolutional approaches based on voxels such as SparseConv (Graham et al., 2018), MinkowkiNet (Choy et al., 2019), or MinkowskiNet+RetroFPN (Xiang et al., 2023). When compared to recent transformer-based architectures such as Stratified Transformer (Lai et al., 2022), PointConvFormer (Wu et al., 2023), Point Transformer v1 (Zhao et al., 2021) and v2 (Wu et al., 2022), Fast Point Transformer (Park et al., 2022), or OctFormer (Wang, 2023), we can see that our model also surpasses most of these methods, only outperformed by Point Transformer v2 (Wu et al., 2022) and OctFormer (Wang, 2023) on the validation set, and by OctFormer (Wang, 2023) on the test set. These results show that using a well-designed PNE with a simple convolution can achieve state-of-the-art results, improving over some of the recent more complex operations based on attention. However, as we show in the next section, the findings presented in this paper can be used to improve these complex operations too.

Tbl. 3 presents the results of our model on the different tasks of the ScanObjNN data set. We can see that our model, despite using a simple linear combination of the embedding dimensions as convolution, significantly outperforms existing methods such as MVTN (Hamdi et al., 2021) on all of the tasks, and even recent architectures such as PointMLP (Ma et al., 2022), the recent PointNeXt (Qian et al., 2022), or even pre-trained models such as P2P-HorNet (Wang et al., 2022).

## 5.7 IMPROVING OTHER ARCHITECTURES

Lastly, we evaluate how the recommendations of this paper can improve existing and more complex convolution operations and architectures. We select the recent PointTransformerv2 architecture (Wu et al., 2022). In the attention modules of this architecture, the operation includes relative positional information of the neighboring points to modulate the attention. This relative position is encoded with an MLP with hidden neurons equal to the number of features of the layer and a ReLU activation function. Following the recommendations listed in this paper, we substitute this embedding with an

Table 2: Mean IoU on the test and validation sets of ScanNet V2.

| Method | Input | Res. | Val. | Test |
|---|---|---|---|---|
| PointNet++ | points | – | 53.5 | 55.7 |
| PointConv | points | – | 61.0 | 66.6 |
| KPConv | points | 4cm | 69.2 | 68.4 |
| SparseConv | voxel | 2cm | 69.3 | 72.5 |
| Point Transf. | points | – | 70.6 | – |
| PointMetaBase | points | – | 72.8 | 71.4 |
| Fast Point Transf. | points | – | 72.1 | – |
| MinkowskiNet | voxel | 2cm | 72.2 | 73.6 |
| + RetroFPN | voxel | 2cm | 74.0 | 74.4 |
| Stratified Transf. | points | 2cm | 74.3 | 73.7 |
| PointConvFormer | points | 2cm | 74.5 | 74.9 |
| Point Transf. V2 | points | 2cm | 75.4 | 75.2 |
| OctFormer | voxels | 1cm | **75.7** | **76.6** |
| Ours | points | 2cm | 74.9 | 75.5 |

Table 3: Results on the test sets of the ScanOb-jNN dataset.

| Method | OBJ_BG | PB_T50_RS |
|---|---|---|
| MVTN | 92.6 | 82.8 |
| PointMLP | – | 85.4 |
| PointNeXt | – | 87.7 |
| P2P-HorNet | – | 89.3 |
| Ours | **92.9** | **90.4** |

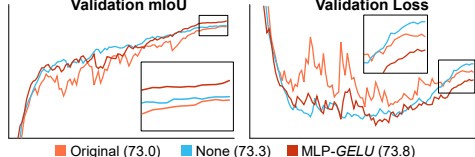

Figure 4: Validation mIoU and loss curves of PointTransformerv2 using different PNE.

MLP with a differentiable activation function, *GELU*, and an embedding dimension of 16. Moreover, we also evaluate the performance of the model if no embedding is used and the attention is modulated by a simple linear combination of the point coordinates. Fig. 4 shows the results of this experiment for the validation set of the ScanNetV2 dataset, with a sampling resolution of 5 cm. First, we can see that by not using any embedding the performance of the model slightly improves, supporting the findings of our previous experiments and indicating that an MLP with a ReLU activation function is usually not adequate to process the relative position of neighboring points. Moreover, we can see that the improved embedding (MLP-*GELU*) provides an increase in performance over the standard architecture. More importantly, Fig. 4 shows that both, not using an embedding and MLP-*GELU*, result in a more stable training, obtaining better mIoU and, in the case of MLP-*GELU*, less over-fitting.

## 6    LIMITATIONS

Although the recommendations and findings of this paper can help to improve existing architectures or to design new ones, the behavior of these might vary depending on each specific operation or architecture. Therefore, even though these recommendations can serve as a good starting point for the initial steps of the network design, all embedding types should be tested.

## 7    CONCLUSIONS

In this paper, we have shown that neighborhood embeddings are a key component in the design of learning architectures for point clouds. Moreover, we have presented the first extensive evaluation of these embeddings in a controlled experimental setup. From these experiments, we derive a set of best practices to help future designs of convolution operations or neural network architectures for point clouds. These recommendations contradict established design choices such as the use of MLP embeddings with *ReLU* activation functions or kNN as a neighborhood selection method. Furthermore, we have shown that an architecture based on a simple convolution that uses such improved embeddings is able to achieve state-of-the-art results on several downstream tasks, outperforming most of the existing methods. Lastly, we have shown how our recommendations can be used to improve existing more complex convolution operations such as PointTransformerv2 (Wu et al., 2022).

We hope our work improves the development of future architectures for point clouds and is able to inspire future research in the development of learning operations for point clouds or further improve existing ones.

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

# A  NETWORK ARCHITECTURE

In this section, we describe the main components of the proposed neural network architecture used in our experiments, see Fig. 5 for an overview.

## A.1  ENCODER

The encoder takes as input the point cloud and computes five down-scaled versions of the point cloud using the *Cell Average* (CA) method (Thomas et al., 2019). The first down-scaling is performed based on a hyper-parameter of the network that defines the size of the voxel cells used in the CA algorithm, $d$. For the four remaining downscale operations, the cell size is computed by doubling the cell size from the previous level. Then, the initial features are computed using a simple linear layer. After, a set of Metaformer blocks (Yu et al., 2022) are used to transform the initial features before transferring them to the next down-scaled point cloud using a convolution operation. This process is repeated until we reach the last down-scaled point cloud. For a classification task, the average of the point features is computed and passed through a linear layer to perform the final prediction. For a segmentation task, the resulting features of each level are used as input by the decoder.

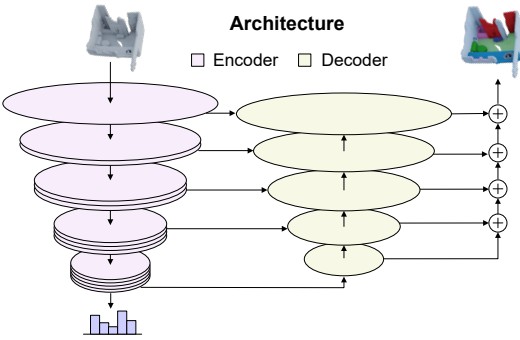

Figure 5: Architecture of the model used in our experiments. The encoder processes the input point cloud using a set of Metaformer blocks (Yu et al., 2022). The process is repeated for different resolutions. For classification, the output of the encoder is processed by a linear layer. For segmentation, a decoder up-scales the features of the encoder.

**Metaformer Blocks.**   Recent work (Yu et al., 2022) has shown that architectural blocks are the driving force behind the success of vision transformer architectures and not the attention modules. Therefore, we adopt this block design in our architecture and substitute the attention module of transformers with our point convolution. Each block is composed of two residual blocks. The first residual block computes feature updates using a point convolution operation. The second residual block computes feature updates using a point-wise MLP with two layers in which the first layer doubles the number of features and the second one reduces it to the desired number of outputs.

**Patch Encoder.**   For the task of semantic segmentation, small cell size usually increases the performance of the model since increases the number of points used by the model. However, these also increase the computational burden of the model. Therefore, in order to take advantage of small cell sizes keeping a small computational cost, we use a patch encoder before the main encoder network, similar to the patch encoder used in vision transformer architectures (Dosovitskiy et al., 2021). This patch encoder extracts features with four convolutional layers from two additional point levels computed using smaller cell sizes.

## A.2  DECODER

The decoder takes as input the feature maps for each of the down-scaled point clouds from the encoder and up-samples the features to the output point cloud for which it performs a final prediction. Our decoder follows a similar architecture as the one proposed by  Kirillov et al. (2019). First, we perform a progressive up-sampling using our point convolutions from the lowest level until the first down-scaled point cloud. Moreover, we use skip connections by summing features from the encoder and decoder to improve information and gradient flow. This results in a feature map for each down-scaled point cloud. Then, we up-sample each feature map to the first down-scaled point cloud using a single up-sampling operation. The five resulting feature maps are then summed together to create the final feature map. A final convolution up-samples these features to the final positions for which we want to perform a prediction and are processed by a single-layer MLP.

## B    ADDITIONAL EXPERIMENTS

### B.1    OBJECT DETECTION ON SCANNET

We, additionally, evaluate all PNE on the task of object detection on the ScanNetV2 dataset. The dataset contains instances of objects from 18 different types of objects. The task consists of predicting the bounding box and object class for each instance. Performance is measured on *Average Precision* (AP) with two different IoU thresholds, 0.25 and 0.5.

**Results.**    In this task, we can see similar results as the one reported for the tasks of classification and semantic segmentation. KP embeddings provide an increased performance when compared to MLP-based embeddings. Moreover, the continuous correlation function *Gauss* performs better in all cases compared to *Trian* for KP embeddings. For the MLP embeddings, contrary to other tasks, *ReLU* activation function performs almost equally well as *GELU* activation function and *Sin* under-performs in this task. Lastly, when we compare neighborhood selection, BQ still provides slightly better results than kNN as experienced in other tasks.

### B.2    SOFTMAX ACTIVATION FUNCTION

Other convolution operations have proposed to use a score function as point neighborhood embedding (Xu et al., 2021). This score function is composed of an MLP followed by a *Softmax* function. In this experiment, we evaluate the viability of *Softmax* as a possible activation for the MLP-based embeddings.

Table 4: Comparison of PNE on the task of object detection on ScanNetV2.

| Neigh. | Emb. | Type | Obj Det. ScanNetV2 | |
| --- | --- | --- | --- | --- |
| | | | AP@25 | AP@50 |
| BQ | KP | Box | 60.6 | 40.9 |
| | | Trian | 61.1 | 42.1 |
| | | Gauss | **62.7** | **42.2** |
| | MLP | ReLU | 58.1 | 37.0 |
| | | GELU | 58.3 | 38.0 |
| | | Sin | 56.5 | 33.9 |
| | None | | 56.5 | 34.4 |
| KNN | KP | Box | 61.3 | 40.2 |
| | | Trian | 62.0 | 40.0 |
| | | Gauss | **62.2** | **42.2** |
| | MLP | ReLU | 56.2 | 33.7 |
| | | GELU | 55.9 | 33.6 |
| | | Sin | 55.3 | 33.2 |
| | None | | 52.5 | 28.3 |

**Results.**    Tbl. 5 presents the results of this experiment. We can see that an MLP-based embedding with a *Softmax* activation function does not provide an improvement over all the other activation functions tested. However, in the future, further analysis of other design decisions of the PAConv operation, such as the aggregation method (MAX instead of SUM) or the addition of global coordinates as input to the embedding function might result in a better performance when combined with Softmax.

## C    EXPERIMENTAL SETUP

In our experiments, we used different experimental setups for each dataset. However, for both datasets, we use AdamW (Loshchilov & Hutter, 2019) optimizer and a OneCycleLR learning rate scheduler (Smith & Topin, 2017) with a maximum learning rate of 0.005, an initial division factor of 10, and a final factor of 1000. We use a weight decay value of $1^{-4}$ and clip the gradient's norm to 100. Moreover, we drop residual paths based on the depth of the layer (Larsson et al., 2017) with a maximum drop rate of 0.5. The final results of a run in our main experiments are computed by accumulating the predictions of the last five saved models. The reported results are the average and stddev of the performance over three different runs.

**Classificaiton on ScanObjNN.**    For the ScanObjNN data set, we use the encoder described in SubSec. A.1 with a number of features for each level equal to $[32, 64, 128, 256, 512]$, a number of blocks per level equal to $[2, 3, 4, 6, 4]$, and an initial grid resolution of $d = 0.05$. During training, we use a batch size equal to 16 and use several data augmentation techniques to transform our input point cloud: random rotation, mirror, random scale, elastic distortion (Nekrasov et al., 2021), jitter coordinates, random adjustments of brightness and contrast of the point's colors, and RGB shift. During the evaluation, we perform a voting strategy with 30 test steps where logits are accumulated

Table 5: Results for the *Softmax* activation function when compared to other MLP designs.

| Neigh. | Task | Metric | MLP | | | |
|---|---|---|---|---|---|---|
| | | | Softmax | ReLU | GELU | Sin |
| BQ | Class. | Acc | 89.8 | 91.1 | **92.8** | 91.9 |
| | | mAcc | 88.9 | 89.6 | **91.4** | 90.6 |
| | Seg. | mIoU | 70.8 | 71.4 | 71.9 | **72.4** |
| | | mAcc | 78.6 | 79.4 | 80.0 | **80.1** |
| | Det. | AP@25 | 52.5 | 58.1 | **58.3** | 56.5 |
| | | AP@50 | 29.4 | 37.0 | **38.0** | 33.9 |
| KNN | Class. | Acc | 90.1 | 89.9 | 91.0 | **92.2** |
| | | mAcc | 88.9 | 88.0 | 89.8 | **90.6** |
| | Seg. | mIoU | 71.6 | 71.0 | 71.0 | **72.1** |
| | | mAcc | 79.6 | 78.9 | 79.1 | **79.8** |
| | Det. | AP@25 | 51.6 | **56.2** | 55.9 | 55.3 |
| | | AP@50 | 29.7 | **33.7** | 33.6 | 33.2 |

over different rotations over the up vector. All models were trained for $400$ epochs on a machine with a single Nvidia $A6000$.

**Semantic segmentation on ScanNetV2.**  For the task of semantic segmentation on ScanNetV2 data set, we use an encoder-decoder architecture as described in Sec. A with a number of features for each level equal to $[64, 128, 192, 256, 320]$, a number of blocks per level equal to $[2, 3, 4, 6, 4]$, and an initial grid resolution of $d = 0.1\,m$. We use the same number of features in the decoder as in the encoder, and $128$ features for the last upsample from all point cloud resolutions. During training, we select rooms until we fill the batch budget of $500\,K$ points, which usually results in $4 - 6$ rooms per batch. Moreover, we use several data augmentation techniques to transform our input point cloud: random rotation, mirror, elastic distortion (Nekrasov et al., 2021), random scale, translation, jitter coordinates, random crop, random adjustments of brightness and contrast of the point's colors, RGB shift, and RGB jitter. Moreover, we mix 2 scenes inside the batch (Nekrasov et al., 2021) with a probability of $0.5$. During the evaluation, as in the classification tasks, we use a voting strategy over different rotations over the up vector. All models were trained for $600$ epochs with $300$ steps each on a machine with a single Nvidia $A6000$.

For comparison to state-of-the-art methods, we increase the cell size of the initial subsample of the encoder to $d = 0.04\,m$. Moreover, we add a patch embedding module before the encoder that processes two additional grid subsamples with resolutions of $d = 0.02\,m$ and $d = 0.03\,m$ with four convolution layers to compute the initial features for the $d = 0.04\,m$ grid. For the final predictions on the test set, we use an over-segmentation method as in Nekrasov et al. (2021).

**Object detection on ScanNetV2.**  For the task of object detection on ScanNetV2 data set, we use as our backbone the same architecture as for semantic segmentation but reduce the number of blocks per level to 2. Our detector is a single-stage detector following the designs of Rukhovich et al. (2022) but with an initial grid resolution of $d = 0.1\,m$ instead of the $d = 0.01\,m$ used in the paper. This allows for faster training and inference at the cost of reduced AP. We implemented our detector using the MMDetection3D (Contributors, 2020) framework and used the standard hyperparameters for a FCAF3D detector: We train for 12 epochs with a batch equal to 8 and an initial learning rate of $0.001$ scaled by $0.1$ after 8 and 11 epochs.

