# OpenReview forum: "Point Neighborhood Embeddings"
_ICLR.cc/2024/Conference — Submitted to ICLR 2024_

### Official Review · Reviewer_BoSJ · 2023-10-30

**Soundness:** 3 good
**Presentation:** 3 good
**Contribution:** 4 excellent
**Rating:** 6
**Confidence:** 5

**Summary:**

This paper gives a comprehensive study on a variety of existing point neighborhood embedding in a controlled setting. Accordingly, it concludes several practical suggestions regarding designing the neighborhood embedding. The paper validates it’s suggestions in the recent point transformer and show the improvement.

**Strengths:**

The paper provides a comprehensive and extremely insightful investigation about the point aggregation module -- the core in point cloud architecture where, given a target point, how we aggregate information from other source points.

This paper fills a blank of this point convolution area -- i.e.Which type of point aggregation is the best? From my viewpoint, most existing works on point cloud architecture like PointNet++ PointConv and KPConv are trying to improve this convolution module in terms of different subtle details like neighborhood querying, kernel function etc. Even the recent point transformer falls in this scope by replacing conv-based aggregation with attention-based way under a mild assumption that the feature of target point is known whereas the point conv doesn’t require it.  Although those works gradually improve the performance on the common benchmark, comprehensive study is still missing -- in other words, different modules are not compared in a controlled setting -- and leads to the best design choice unknown. Because of this, I’ve been bothered so many times by having no idea about which modules are better. So from my perspective, this paper fixes this problem in this area and I believe it can inspire future works like mine.

The derived suggestions are very useful. It includes: no ReLU in MLP, ball-query better KNN and kernel point is better than MLP. For the latter two, I think it’s not surprising as I also observe this. But it’s still useful to have a fair comparison to show it. And for the first one, it’s something new to me.

Also, from my viewpoint,  the formula presented by the author is easy to follow. I would consider using the same formula -- it summarizes the most of existing convolution in a unified way-- when I teach a lecture about point convolution next time.

**Weaknesses:**

The paper has several weaknesses which I’ll detail below.

More modules could be considered -- for example like the very powerful PAConv that defines the convex kernel function. Also the recent PointMLP [1] defines an interesting local aggregation module that relies on normalization. I think the author can further improve the paper by including more options.

The paper utilizes the KPConv with fixed kernel point location which is suboptimal in original KPConv paper. So I would recommend discussing how the deformable KPConv fits in the formula. This would make the paper more comprehensive.

I’m not very sure if I can agree with the argument made in Eq. 1 where the Point transformer is not a convolution. From my perspective,  it also has a similar formula if we consider the query point’s features. But this is my personal viewpoint. I wouldn’t convince authors to convey point transformers in a convolution way. However, given that the author doesn’t consider the point transformer as a kind of convolution, it’s a bit weird to validate the proposed tips derived from convolution into the point transformer. So I think the author might want to clarify it a bit in the paper.

The paper didn’t use the ModelNet40. Although it’s a synthetic dataset that is highly saturated by lots of existing methods, I believe that it’s still a proper dataset to fairly investigate the different subtle modules in point convolution. Or the paper needs to justify why the paper doesn’t use the popular ModelNet40 in my opinion.

[1]Ma, Xu, et al. "Rethinking network design and local geometry in point cloud: A simple residual MLP framework."  ICLR 22.

**Questions:**

Please address the questions above

---

> ### Author Response · Authors · 2023-11-22
> **Additional activation function and discussions**
>
> We would like to thank the reviewer for appreciating the relevance of our work and for all the suggestions. In the following paragraphs, we would like to address the main concerns of the reviewer:
>
> **Including more options such as PAConv designs**
> >We have experimented with other MLP designs by including the Softmax activation function in our MLP-based embeddings in a similar way as done in the Score module for PAConv.
> >The results of this experiment can be seen in the following table:
>
> |  Neigh | Task  | Metric  | Softmax | ReLU | GELU | Sin |
> |---|---|---|---|---|---|---|
> |  BQ |  Class |  Acc. | 89.8 | 91.1 | **92.8** | 91.9 |
> |   |  |  mAcc | 88.9 | 89.6 | **91.4** | 90.6 |
> ||||||||
> |   | Seg. |  mIoU | 70.8 | 71.4 | 71.9 | **72.4** |
> |   |  |  mAcc | 78.6 | 79.4 | 80.0 | **80.1** |
> ||||||||
> |   | Det. |  AP@25 | 52.5 | 58.1 | **58.3** | 56.5 |
> |   |  |  AP@50 | 29.5 | 37.0 | **38.0** | 33.9 |
> ||||||||
> | KNN  | Class. |  Acc | 90.1 | 89.9 | 91.0 | **92.2** |
> |   |  |  mAcc | 88.9 | 88.0 | 89.8 | **90.6** |
> ||||||||
> |   | Seg. |  mIoU | 71.6 | 71.0 | 71.0 | **72.1** |
> |   |      |  mAcc | 79.6 | 78.9 | 79.1 | **79.8** |
> ||||||||
> |   | Det. | AP@25 | 51.6 | **56.2** | 55.9 | 55.3 |
> |   |      | AP@50 | 29.7 | **33.7** | 33.6 | 33.2 |
>
> >We can see that an MLP-based embedding with a Softmax activation function, unfortunately, does not provide an improvement over the other activation functions tested.
> >However, in the future, it will be interesting to analyze other design decisions of the PAConv operation, such as the aggregation method (MAX instead of SUM) or the addition of global coordinates as input to the embedding function, which might result in a better performance when combined with Softmax.
>
> **Deformable KPConv**
> >We thank the reviewer for pointing this out, we have included a discussion regarding deformable kernel points in the paper
>
> **Transformer is not a convolution**
> >We regret we did not communicate this properly.
> >We agree with the reviewer that PointTrasnformer is a convolution since it aggregates information from a local neighborhood using the relative position between points.
> >Our statements after Eq. (1) intended to describe that these operations, such as PointTransformer, cannot be defined with Eq(1) as we presented it, but this did not mean that our Eq(1) can represent all possible definitions of convolution.
> >We have made this point clearer in the revised version of the paper.
>
> **Validation on ModelNet40**
> >We have chosen to validate on ScanObjNN because of the problems pointed out by the reviewer, performance is saturated by many models and it might be difficult to measure the improvements between embeddings.
> >Therefore, due to the limited time of the rebuttal phase and as suggested by another reviewer, we have chosen instead to incorporate the task of object detection on ScanNet to increase our evaluation.

---

> > ### Comment · Reviewer_BoSJ · 2023-11-22
> > **Thanks for the feedbacks**
> >
> > Thanks for the feedback and sorry for the late.
> > I think rebuttal addressed most of my concerns. And I'll adjust my rating combining comments from other reviewers.

---

### Official Review · Reviewer_HsUz · 2023-10-31

**Soundness:** 3 good
**Presentation:** 3 good
**Contribution:** 3 good
**Rating:** 8
**Confidence:** 3

**Summary:**

This paper presents a comprehensive study in the performances of various Point Neighborhood Embeddings (PNE) mechanisms in point convolutional neural network architectures, and further offers recommendations for improving model designs based on the findings. They validated that their recommendations can outperform most existing methods in several tasks with simple design and can improve existing complex convolution operations.

**Strengths:**

1. Their findings and recommendations can benefit future arch design of point clouds
2. They did comprehensive experiments to explore the different design choices and validate their recommendations.
3. The writing is good with detailed introduction and analysis.

**Weaknesses:**

-

**Questions:**

-

---

> ### Author Response · Authors · 2023-11-22
> **We thank the reviewer for possitive assesment**
>
> We thank the reviewer for the positive assessment of our work. We also hope that our work, as indicated by the reviewer, can benefit future architectural designs of point clouds.

---

### Official Review · Reviewer_n5A1 · 2023-11-01

**Soundness:** 2 fair
**Presentation:** 3 good
**Contribution:** 1 poor
**Rating:** 3
**Confidence:** 4

**Summary:**

The paper presents extensive study that analyzes point cloud embeddings based on activation functions used, correlation functions used, MLP vs Kernel points embeddings, etc. The paper also talks about different convolution operation and the neighbourhood election based on ball query vs k-NN. The authors performed two downstream tasks classification and segmentation on two benchmark datasets.

**Strengths:**

1. The paper is well-written and provides a good analysis of point cloud embeddings.
2. This can help build new algorithms/architectures to improve embeddings.
3. The study provides interesting results.
4. The architecture includes simple modification and not expensive operations like transformers.

**Weaknesses:**

1. This paper can be seen as a good experimental study paper which is not up to the level of ICLR. This is like a review paper although in a different direction where it provides extensive study on multiple points.
2. The paper is very weak in novelty and makes some claims without evidence or explanations except results.
3. The work mentions a lot of comparisons between activation functions, MLP vs. KP, etc. However, the whole paper lacks in explaining “why” something is better or worse. For example, kNN vs BQ talks about having high variance with kNN. It does not explain why.
4. Support of the embedding is not clearly defined. I believe the receptive field is neighbourhood.
5. There is no fixed proposed architecture in the paper. The results are not generalized based on a particular setting. The architecture used is the existing architecture mentioned in 5.1 with some changes like activation function, using KP/MLP, and different correlation function based on KP embeddings.

**Questions:**

Why validation set is used for PNE and the validation+test set is used for other methods?

---

> ### Author Response · Authors · 2023-11-22
> **Addressing main concerns**
>
> We thank the reviewer for the feedback and hope we can address the main concerns in the following paragraphs.
>
> **Weak novelty and not up to the level of ICLR**
> >Novelty is a very subjective matter, and it can be difficult to rebut.
> >However, we would like to point out that no other reviewer had this concern, and provided instead encouraging remarks such as "The paper provides a comprehensive and extremely insightful investigation about the point aggregation module.", "This paper fills a blank of this point convolution area", or "Their findings and recommendations can benefit future arch design of point clouds".
> >We believe our work goes far beyond a simple review paper and provides several findings that can help move forward the field of learning on point clouds.
>
>
> **kNN vs BQ talks about having high variance with kNN by does not explain "why"**
> >Unfortunately, we were not able to communicate this properly.
> >The reason why the high variance might hurt performance was described in the paragraph kNN vs BQ.
> >Large variance will result in a large embedding norm for some MLP-based embeddings, ReLU and GELU, making training difficult.
> >Moreover, large variance will also result in some outlier points being not part of the support for some KP-based embeddings, *Gauss* and *Trian*, making them irrelevant for these computations.
> >We have updated this paragraph to make the point clearer.
>
>
> **Support of the embedding is not clearly defined. I believe the receptive field is neighbourhood.**
> >We regret that this was not clear in the paper.
> >The support of the embedding is the same as the support in any function: " subset of the function domain containing the elements which are not mapped to zero".
> >
> >The receptive field of the convolution for point clouds is the same as for discrete convolutions, the area around the point that affects the resulting feature computation.
> >This term is adopted from neuroscience where it is defined as: "The receptive field of an individual sensory neuron is the particular region of the sensory space (e.g., the body surface, or the visual field) in which a stimulus will modify the firing of that neuron. "
>
>
> **There is no fixed proposed architecture in the paper.**
> >Regretfully, we did not understand this specific concern of the reviewer.
> > We believe the reviewer meant that our architecture is not original.
> >However, this was not the aim of the paper.
> >We aim to analyze and compare existing convolution designs from which we were able to gather new insights that help future designs of architectures for point clouds.
>
>
> **Why validation set is used for PNE and the validation+test set is used for other methods?**
> >The labels for the test set of ScanNet are hidden and one is required to upload the predictions to their system.
> >The system only allows uploading a set of predictions once every two weeks and only allows for a method per institution.
> >Therefore, it was impossible for us to perform the PNE comparison on the test set.

---

> > ### Comment · Reviewer_n5A1 · 2023-11-23
> > **Thanks**
> >
> > Thanks for the feedback. I went through the answers to my concerns. I will consolidate these in the final comments and rating.

---

### Official Review · Reviewer_1sX7 · 2023-11-01

**Soundness:** 3 good
**Presentation:** 3 good
**Contribution:** 2 fair
**Rating:** 5
**Confidence:** 3

**Summary:**

This paper discusses different types of embeddings and aggregation methods for neighboring 3D point clouds. By experimenting with different combinations on ScanObjNN and ScanNet, the author summarizes a set of best practices for designing new PNE.

**Strengths:**

1. The review of previous 3D point cloud embeddings is comprehensive and logical.

2. The experiments and analysis on two tasks (classification and segmentation) are careful and in-depth.

**Weaknesses:**

The improvement is not obvious compared to existing methods in Table 2 compared to 3. Does this mean the effect of PNE is less important when training data is less? I recommend to experiment on larger-scale 3D datasets for classification, such as Objverse, or other 3D tasks, such as indoor/outdoor 3D object detection. The two experiments on the paper is not sufficient enough to demonstrate the conclusion.

**Questions:**

One related work investigating Sin for 3D classification is expected to be included and discussed: *Starting from Non-Parametric Networks for 3D Point Cloud Analysis* accepted by CVPR 2023.

---

> ### Author Response · Authors · 2023-11-22
> **New object detection task**
>
> We would like to thank the reviewer for the positive feedback and suggestions. In the following paragraphs, we would like to address the main concerns of the reviewer:
>
> **Improved not clear on Table 2**
> >We regret we were not able to communicate this point clearly.
> >In this paper, we do not aim to propose a new architecture or convolution to improve over existing methods.
> >We show instead that simple convolution for point clouds can provide competitive performance when compared to more recent and *complex* operations such as transformer architecture which use self-attention in their layers.
> >However, when compared to other architectures without self-attention, our method outperforms those by a significant margin (MinkowskiNet and lower).
>
>
> **Other 3D tasks**
> >Following the suggestion of the reviewer, we have trained our model with all PNE used in the paper on the task of object detection on the ScanNet dataset.
> >The results of this experiment are presented in the following table:
>
> |  Neigh | Emb.  | Type  |AP@25|AP@50|
> |---|---|---|---|---|
> |  BQ |  KP |  Box | 60.6  |  40.9 |
> |   |   | Trian  | 61.1  | 42.1  |
> |   |   | Gauss  |  **62.7** |  **42.2** |
> | | | | | |
> |   | MLP  | ReLU  | 58.1  | 37.0  |
> |   |   | GELU  |  **58.3** |  **38.0** |
> |   |   |  Sin |  56.5 |  33.9 |
> | | | | | |
> |   |  None |   |  56.5 | 34.4  |
> | | | | | |
> |  KNN | KP  | Box  | 61.3 |  40.2  |
> |   |   | Trian  | 62.0 |  40.0  |
> |   |   |  Gauss | **62.2** |  **42.2**  |
> | | | | | |
> |   |  MLP |  ReLU |  **56.2** |  **33.7**  |
> |   |   |  GELU | 55.9 |  33.6S  |
> |   |   |  Sin | 55.3 |  33.2  |
> | | | | | |
> |   | None  |   | 52.5 |  28.3  |
>
> >In this task, we can see the same trends reported for the tasks of classification and semantic segmentation.
> >KP embeddings provide an increased performance when compared to MLP-based embeddings.
> >Moreover, the continuous correlation function *Gauss* performs better in all cases compared to *Trian* for KP embeddings.
> >For the MLP embeddings, contrary to other tasks, *ReLU* activation function performs almost equally well as *GELU* activation function, and *Sin* under-performs.
> >Lastly, when we compare neighborhood selection, BQ still provides slightly better results than kNN as experienced in other tasks.
> >We have included such experiments in the Appendix.
>
> **Related work: Starting from Non-Parametric Networks for 3D Point Cloud Analysis**
> > We thank the reviewer for pointing out this relevant work.
> >We have included this reference when describing the sin activation function.

---

### Author Response · Authors · 2023-11-22
**Revised version of the paper**

We would like to thank the reviewers for all the positive feedback, and the insightful comments and recommendations that have helped improve our paper.
Moreover, we believe that remarks from the reviewers such as *The paper provides a comprehensive and extremely insightful investigation about the point aggregation module.*, *This paper fills a blank of this point convolution area*, or *Their findings and recommendations can benefit future arch design of point clouds*, suggest that our work can be of high relevance for a prestigious conference such as ICLR.

Following the suggestion of the reviewers, we have uploaded a new revised version of the paper.
However, we additionally try to answer each reviewer individually here.
A summary of the main changes in the paper is listed below:

* Addition of the object detection task on ScanNet
* Addition of new activation function used in PAConv, Softmax.
* Added discussion of Adaptative Kernel Point locations.
* Clarify parts of the text: kNN vs BQ, and PointTransformer as convolution.

---

### Meta-Review · Area_Chair_5zLA · 2023-12-05

**Metareview:**

The authors have presented an observation study on point neighborhood embedding (PNE) in the context of point cloud neural network architectures for encoding neighborhood point information in 3D space. The major finding of the paper is that simple methods (e.g., linear combinations of point coordinates) can outperform the most commonly used embedding technique, using multi-layer perceptrons (MLP) with ReLU activations. After the discussion period, two reviewers gave rejection, and two reviewers gave acceptance. The main concern of the reviewers is that the observations made by the authors are not clearly explained and require more in-depth analysis. While the authors provided explanations of the performance difference based on the variance of the activation norm (which may unstabilize the training), AC also thinks that such a claim should be more rigorously verified. For instance, sinusoidal activation may have lower variance than others (as the norm is bounded) but does not always perform better. Furthermore, AC also agrees with the reviewers that more evaluation datasets are required than the current two datasets to rigorously verify the consistent observation, which AC thinks is an important component of observation studies. Overall, AC tends to recommend rejection.

**Justification For Why Not Higher Score:**

AC also carefully read this paper, rebuttal, and the discussion between authors and reviewers. However AC also generally agrees with the concerns of reviewers. The paper needs more rigorous analysis to give explanations why other embedding techniques outperform conventional embedding and should verify a consistent observation across datasets by considering multiple datasets, e.g., MobileNet40.

**Justification For Why Not Lower Score:**

N/A

---

### Decision · Program_Chairs · 2024-01-16

Reject